# Pilates to Improve Core Muscle Activation in Chronic Low Back Pain: A Systematic Review

**DOI:** 10.3390/healthcare11101404

**Published:** 2023-05-12

**Authors:** Jennifer Franks, Claire Thwaites, Meg E. Morris

**Affiliations:** 1Academic and Research Collaborative in Health (ARCH), La Trobe University, Melbourne, VIC 3086, Australia; j.franks@latrobe.edu.au (J.F.); c.thwaites@latrobe.edu.au (C.T.); 2Victorian Rehabilitation Centre, Healthscope, Glen Waverley, VIC 3150, Australia

**Keywords:** low back pain, Pilates, exercise, core muscle activation, core muscle strength, pain, disability

## Abstract

Low back pain is prevalent in the community and associated with deficits in core muscle strength and activation. Pilates is argued to improve movement and reduce pain, yet there is a limited understanding of the specific effects of Pilates training on core muscle strength or activity. A systematic search of databases (CINAHL, Embase, Ovid MEDLINE) used Preferred Reporting Items for Systematic Reviews and Meta-Analyses (PRISMA) methods to evaluate randomised controlled trials (RCTs) on the effects of Pilates to improve core muscle activation. Methodological quality was assessed using the Physiotherapy Evidence Database scale (PEDro). The certainty of findings was determined using the Grading of Recommendations Assessment, Development and Evaluation tool. Of the initial yield of 563 articles, eight RCTs met the inclusion criteria. A diverse range of Pilates interventions and outcome measures were utilised to assess effects on core muscle activation and strength. The main finding was that Pilates is not inferior to equivalently dosed exercises, and can be superior to non-equivalent or no exercise, for improving core muscle strength as indicated by muscle thickness. There was emerging evidence that Pilates positively impacts core muscle strength and can be an effective intervention for people living with chronic low back pain.

## 1. Introduction

Pilates is gaining popularity worldwide as a therapeutic intervention to improve movement and to reduce pain [1,2,3]. Pilates has evolved from a training method within the dance community [4,5,6] to a therapeutic intervention for the management of low back pain [7,8] and other conditions in the general community [9,10]. Around 80% of people experience low back pain at least once during their lifetime [11,12], with prevalence increasing with age [13,14] and occurring more frequently in women [13,15,16]. People with low back pain are known to demonstrate inhibition and altered recruitment of local core muscles that ordinarily contract prior to trunk muscles, such as erector spinae and rectus abdominis, to maintain proximal trunk stability [17,18,19,20]. Local core muscles include transversus abdominis, lumbar multifidus and internal obliques, and form a muscle system with the pelvic floor and diaphragm to stabilise the trunk prior to limb movement [17,21].

Pilates is arguably an effective intervention to reduce low back pain [7,22,23]. There is emerging evidence in some people that Pilates training is associated with changes in muscle strength, muscle recruitment or muscle activation [24,25,26,27]. There are also promising results from several recent systematic reviews on Pilates for low back pain, with low- to moderate-quality evidence supporting short-term improvements in pain [7,28,29], kinesiophobia [30], and disability [28,29]. Nevertheless, there remains a limited understanding of mechanisms of effect or quantifiable effects of Pilates in transforming core muscle strength in people with chronic low back pain. It is important to understand whether Pilates can elicit these specific changes to establish its efficacy as a rehabilitation intervention for people with this persistent, debilitating condition.

As a popular exercise modality, it is imperative to understand the best available evidence on the validity of Pilates for low back pain, to guide decision making for clinicians who treat people with this debilitating condition. The main aim of this systematic review was to synthesize and evaluate research conducted as randomised controlled trials (RCTs) to answer the question: Is Pilates exercise effective for improving core muscle strength/core muscle activation in people living with chronic low back pain? A secondary aim was to examine the effects of Pilates on pain, disability, and quality of life in people with chronic low back pain.

## 2. Materials and Methods

The methodology of this systematic review is grounded by the Preferred Reporting Items for Systematic Reviews and Meta-Analyses (PRISMA) framework [31] and AMSTAR-2 recommendations [32]. The review was prospectively registered with PROSPERO (CRD42022363368). The PICO format was utilised to develop the search strategy with terms and limits relating to the population of interest and intervention. MeSH headings informed keywords for low back pain, muscles, and Pilates or exercise movement interventions. A university librarian worked with the research team to create and implement the search strategy which included CINAHL, EMBASE, and Ovid MEDLINE from inception until 19 December 2022, with the Ovid MEDLINE example provided in Appendix A. Grey literature was searched through Google Scholar, and hand searching of reference lists of included articles and relevant systematic reviews was undertaken.

The Covidence online platform [33] was accessed to import identified articles and remove duplicates. Titles and abstracts were independently screened by two reviewers, with full texts retrieved to determine eligibility. Whilst all forms of primary and secondary literature were searched, only RCTs were included to reduce potential confounding by selection bias from non-randomised research [34]. Methodological quality was evaluated using the PEDro scale [35], with scores calculated by two authors trained in scale application. The PEDro scale contains 11 items that assess eligibility criteria, random allocation, concealed allocation, similarity at baseline, blinding of participants, therapists, and assessors, greater than 85% retention, intention-to-treat analysis, between-group statistical comparisons and point measures, and measures of variability. Each item can potentially contribute one point to the overall score out of 10. The PEDro has been shown to be a valid measure of methodological quality of clinical trials [36]. All trials were included in the review irrespective of quality score due to the limited amount of research addressing this topic. The scoring for each included trial was tabulated for ease of comparison [37].

Studies were included if participants were adults aged 18 years and over who had low back pain of more than three months’ duration, with no exclusion criteria related to underlying aetiology. Interventions were limited to those with a defined exercise protocol based on the Pilates method, with or without apparatus use. Comparator groups could be any intervention or no intervention. The primary outcomes of interest were prioritised as core muscle thickness as measured by real-time ultrasound imaging or core muscle activation as measured by surface electromyography, pressure biofeedback units, or functional movement tests. Real-time ultrasound performed by trained personnel currently produces the most reliable measure of core muscle thickness at rest and during activity [38]. Methods to measure core muscle activation have sensitivity and accuracy issues associated with isolated use of surface electromyography [39], or pressure biofeedback [40]. Whilst functional movement tests are able to distinguish between those with and without low back pain, there is limited specificity with regards to local core muscle and global trunk muscle contributions [41]. Core muscles were defined to be transversus abdominis, lumbar multifidus, and internal oblique muscles, and distinguished from global trunk muscles [17]. Secondary outcomes of interest were pain, disability, and quality of life measures. Attempts were made to contact the original authors for clarification if published information was insufficient to establish eligibility. Two reviewers independently extracted information about sample and intervention characteristics, outcome results at all collected time points, and participant-reported pain, disability, and quality of life. A third reviewer was used if conflicts could not be resolved. Information was cross-checked to confirm accuracy.

To address the primary aim, data were sought on changes in core muscle thickness (i.e., cross-sectional area at rest and during activity) [42] or changes in core muscle activation (i.e., time to muscle activation onset, duration of muscle contraction, change in activation pressures) [43,44]. Secondary outcome data were sought from reliable and validated patient-reported outcome measurements (PROMS) in low back pain populations across pain, disability, or quality of life (i.e., Oswestry disability index [45,46], visual analogue scale [47], Short Form-36 [48]).

Data pertaining to baseline and post-intervention assessments, within-group changes, and between-group differences in core muscles and PROMS were tabulated then summarised narratively. To interpret variables likely to contribute to heterogeneity, the data were categorised by comparator type (equivalent exercise intervention, non-equivalent exercise intervention, or non-exercise intervention). We defined an equivalent exercise intervention to be bodyweight-resisted strength-focused exercise [49], performed with similar clinical supervision and dosage. Given the broad inclusion criteria, this approach was necessary as we captured diverse study designs; varied type, intensity, duration, and supervision of interventions; low back pain of varying aetiology; and different assessment time points. Where experiment design contained more than two groups, we followed synthesis recommendations [50] and pooled equivalent exercise groups (e.g., mat Pilates and apparatus Pilates) or disregarded outcome measures from groups not relevant to our review question such as those where exercises were replicated within intervention groups, or the third group was a non-exercise comparison.

In addition to insufficient data for statistical pooling, heterogeneity of populations, research design, interventions, and outcomes meant that a meta-analysis was not feasible. A summary of findings was created using GRADE [51], with an evidence table created using GRADEpro [52]. This approach allows for assessment of the overall quality of a body of evidence, which is downgraded or upgraded depending on preset criteria and results in a rating of very low-, low-, moderate-, or high-quality evidence. This method of summarising findings was used where an outcome measurement was utilised by more than one trial. As we only included RCTs, evidence quality started with a high rating. It was then downgraded if most of the trials scored ≤ 6 on the PEDro scale, indicating less than high methodological quality; where there was risk of imprecision due to small sample sizes (<400); where there was potential for indirectness when studies did not compare equivalent interventions; or risk of inconsistency due to heterogeneity across populationsor interventions.

## 3. Results

### 3.1. Search Results

Following removal of duplicates, 563 articles were identified. Of these, 514 were excluded after title and abstract examination. The full-text review excluded a further 41 articles. Two trials [53,54] initially appeared to meet the inclusion criteria; however, closer examination by a third reviewer identified that group allocation was not random and they were subsequently excluded. From this yield of studies, it was not necessary to contact the authors for further detail. Full-text exclusion criteria rationale is provided in Appendix A, and the literature selection process is outlined in Figure 1.

### 3.2. Methodological Quality

The overall quality of trials was “fair”, with a mean PEDro score of 5.9 out of 10 [55]. The trials ranged from three [56] to eight [57]. All reported eligibility criteria, between-group differences, and point estimates, with six trials reporting less than a 15% dropout rate [57,58,59,60,61,62], and five reporting use of blinded assessors [57,59,60,62,63]. See Table 1 for critical appraisal scoring.

### 3.3. Study Characteristics

Of the eight included articles [56,57,58,59,60,61,62,63], all were RCTs published between 2012 and 2022. Studies were conducted in several countries including Iran [58,62], Turkey [59,63], India [56], Spain [60], Australia [57], and Brazil [61].

### 3.4. Participant Characteristics

The number of participants in any single trial ranged from 30 to 98, with ages ranging from 18 to 75 years. There was a total of 437 participants included in the review (222 allocated to an intervention based on a Pilates method, 142 allocated to an alternative exercise, and 73 allocated to no exercise). Of the six trials reporting on participant gender, three [59,61,63] recruited female participants only, and three [56,57,60] recruited a mix of gender. Participant body mass index was reported in six studies [56,58,59,61,62,63], with values in three of these [58,59,61] reporting average participant body mass index to exceed 25.0 kg/m^2^, defined as overweight [65]. Chronic low back pain was defined as symptom duration of more than three months in six studies [56,57,58,59,60,62], and six months duration in a single trial [61]. Further exclusion criteria were detailed as any pathological or traumatic cause of symptoms in three studies [58,60,62]. Trial characteristics are outlined in Table 2 and details of eligibility criteria in Appendix A.

### 3.5. Interventions

Intervention settings were limited to outpatient clinics in four trials [56,58,59,62]. The setting was not described in the remainder. Six trials reported interventions being delivered by physiotherapists [56,59,62], or physiotherapists with additional Pilates training [60,61,63] supervising interventions. When reported, interventions were delivered in groups of up to 12 participants with one practitioner supervising [56,57,59,60,61], or individually [63]. Pilates or Pilates-type stabilisation exercises were consistently described, with several outlining exercise progressions [61,62,63]. Exercise sessions ranged from 50 to 70 min, and frequency from one to five sessions per week over three to 12 weeks. Additional therapies were outlined in four trials in the form of transcutaneous epithelial nerve stimulation [58,62], interferential therapy [56], hot packs [56], and therapeutic ultrasound [58].

Four studies compared Pilates training to groups performing equivalent exercise including dynamic lumbar stabilisation [56], and variations of routine resistance, stretching, cardiovascular and balance exercises [58,61,62]. Three investigations were multi-arm RCTs. One compared a mat-based Pilates group and an apparatus-based Pilates group to a non-exercise control group [60]. Another compared mat Pilates exercises to lumbar stabilisation exercises and a dynamic exercise intervention group [56]. The third comparison was mat Pilates, general exercise, and education alone [61]. One trial compared mat and apparatus Pilates exercises to indoor stationary cycling [57]. Pregnancy and low back pain was examined once, with comparison of Pilates to usual antenatal care without exercise supervision [63]. None of the included studies examined outcomes after the post-intervention assessment.

### 3.6. Muscle Thickness and Muscle Activation

Table 3 summarises the outcomes within the primary domains of muscle thickness and muscle activation. Muscle thickness changes assessed with real-time ultrasound were evaluated in three trials, one with an equivalent exercise comparator [62] that reported significant within-group changes in the Pilates group. An investigation with a non-equivalent comparator found a positive significant between-group change in the Pilates group in the transverse abdominis, lumbar multifidus, and internal oblique muscles [59]. Both within-group and between-group positive changes were reported in a trial of two Pilates interventions compared to a non-exercise group, with pooled results for muscle thickness changes favouring the Pilates groups [60].

Across the trials, muscle activation was examined using pressure biofeedback units three times [56,58,63], and surface electromyography [57,61] or functional movement tests twice [58,59]. The three studies with a Pilates intervention and an equivalent exercise comparator [56,58,61] reported positive within-group differences in muscle activation in the Pilates groups. One reported a significant between-group difference in the activation of the right lumbar multifidus muscle, as measured by surface electromyography, favouring the Pilates intervention [61]. Between-group analysis in two studies did not find a difference in any muscle activation with equivalent interventions [56,58]. The Batibay et al. investigation noted reported a positive yet non-significant change favouring the Pilates intervention [59]. The trial by Brooks et al. [57] did not find between-group differences. The other trial with a non-exercise comparator [63] reported significant between-group differences favouring the Pilates interventions.

### 3.7. Pain, Disability, and Quality of Life

Table 3 summarises the outcome measurement data assessing pain, disability, and quality of life. All trials investigated changes in pain following a Pilates intervention using visual analogue scales. Aside from a single investigation only reporting baseline values [61], all reported positive significant within-group pain reduction scores with Pilates. One with an equivalent exercise comparator [58] noted superior pain reduction in the Pilates intervention. Another reported a small non-significant difference between groups [56]. For trials with non-equivalent or non-exercise comparators, three reported significant within-group and between-group changes in pain reduction favouring Pilates interventions [57,59,63]. One found a positive but non-significant change [60]. Post-intervention pain outcome data for Pilates and comparator groups have been illustrated in a forest plot (see Figure 2). The minimal clinically important difference in use of a visual analogue scale in subacute and chronic low back pain has been reported as 20 mm, or a change of two points [66]. As seen in Figure 2, three studies [57,60,63] with non-equivalent comparator groups reported improvements of at least two points, indicating meaningful improvements in pain.

The range of PROMS used to assess disability are outlined In Table 4, with the Oswestry disability index applied in four trials [56,57,58,63]. All reported significant within-group reductions in disability following Pilates training. In two trials with an equivalent comparator, a positive but non-significant change favouring Pilates was reported once [56], and no difference between groups in the other [58]. The remainder noted between-group disability measures to significantly favour Pilates interventions [57,59,60,63]. Post-intervention data as measured by the Oswestry disability index are illustrated in a forest plot (see Figure 3). Published minimal clinically important difference scores for this PROM in low back pain populations are between 9.5 [67] and 12.8 points [68], with only one of the studies included in this review reporting this magnitude of improvement [56].

Quality of life was assessed in two trials with non-equivalent or non-exercise comparators. One [59] used the Short Form-36 tool and reported a significant between-group change favouring the Pilates group. The other [63] used the Nottingham Health Profile and did not find any differences between groups.

### 3.8. Certainty of Findings

The certainty of findings for all outcome measures used by more than one study was rated very low (Table 4), reflecting the quality and low participant numbers of included studies. Inconsistency, indirectness, and imprecision can be partly attributed to the limited number of RCTs and variance in assessment in this emerging area of research. However, the data tables and syntheses suggest a degree of certainty that Pilates interventions are not inferior to other exercise or non-exercise interventions in people with low back pain. 

## 4. Discussion

The results of this systematic review provide emerging evidence that Pilates can assist some people living with low back pain to increase the strength of their core muscles around the trunk, pelvis, and abdomen. Pilates was not inferior to equivalently dosed exercises, other dosages of exercise, or other physical activities for improving core muscle strength, shown by muscle thickness measured using real-time ultrasound assessment [59,60,62]. Core muscle activation was assessed by a myriad of methods, and where Pilates was compared to an equivalent exercise intervention, surface electromyography results found that Pilates was not inferior [61].

For people with chronic low back pain, Pilates was not inferior to equivalent exercise interventions to reduce pain as assessed by visual analogue scales. Nevertheless, it did appear to be more effective at reducing pain than not exercising in some individuals. The systematic review did not allow inferences to be drawn about quality of life, because few reviewed studies measured this variable.

Our findings agree with the conclusions drawn in the systematic review of muscle activation in people with chronic low back pain following Pilates by Romão et al. [24]. However, the three studies of the Romão review were limited to electromyography assessment, subsequently quantifying muscle activation, not core muscle thickness. The systematic review of the effect of Pilates on pain and disability in chronic nonspecific low back pain by Wong et al. [69] did not identify strong evidence of any preferential exercise type for patients with chronic nonspecific low back pain. In contrast, a meta-analysis on different modes of exercise for chronic nonspecific low back pain [70] reported low-quality evidence that supervised Pilates, alongside other “active therapies” where exercise was guided and progressed and was effective for reducing pain and subjective physical function. This finding concurs with the network meta-analysis by Hayden et al. [71], who reported clinically significant benefits of Pilates compared to other exercises for pain intensity and functional limitations. Like our review, the certainty of findings was limited by within-study risk of bias and heterogeneity.

The findings of our review lend some support to the notion that Pilates can improve health and well-being in people with chronic low back pain. An inherent component of this exercise approach is facilitating the coordinated contraction, or motor control, of the deep muscles, predominantly multifidus, transverse abdominis, the pelvic floor, and diaphragm, that are responsible for spinal stability and reduced joint compression [72]. It is proposed that in people with chronic low back pain, there are motor control impairments with delayed onset of the deep muscles, and subsequent overactivity of superficial muscles compensating for the lack of stability [73]. Pilates exercises aim to address the motor control of the deep muscles and reduce the activity of the superficial muscles, in addition to improving body awareness and posture [28]. Results of this systematic review provide support to this biological rationale, with evidence of strength improvements in the deep muscles observed in several studies. The reported reductions in subjective pain and disability suggest a positive impact of Pilates on well-being in some individuals with this chronic condition.

There were some limitations to this systematic review. Most notably, only eight RCTs were included in the final yield, all of which were reported in English. This limits the generalizability of the findings to different populations, cultures, and geographical regions. The certainty of the review findings was rated as very low, reflecting the generally low quality and small participant numbers for the included studies. Inconsistency, indirectness, and imprecision can be partly attributed to the limited number of RCTs identified. There was a consistent lack of long-term follow-up, similar to other studies of Pilates and chronic low back pain [7,28,29,69]. In this review, we did not consider exclusion criteria related to aetiology, and it is possible that different results in response to Pilates could be found for different medical conditions. Finally, we did not compare Pilates interventions with pharmacological treatments for chronic low back pain, and this would be a valuable topic for future clinical trials.

## 5. Conclusions

There is emerging evidence that Pilates is not inferior to equivalently dosed exercises for improving core muscle strength in people living with chronic low back pain. The RCTs reviewed suggest that Pilates can be more beneficial than not exercising in some individuals, by improving core muscle strength or by reducing pain. With movement and exercise routinely recommended for this condition, Pilates appears to be a valid option for clinicians to consider.

## Figures and Tables

**Figure 1 healthcare-11-01404-f001:**
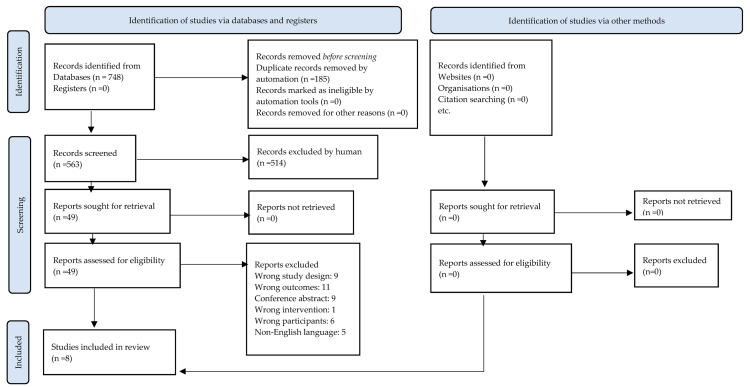
PRISMA flowchart [32] outlining study selection process.

**Figure 2 healthcare-11-01404-f002:**
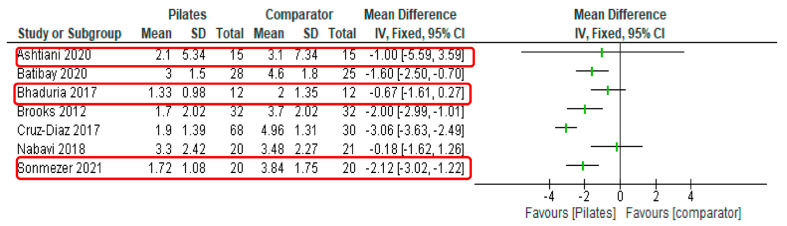
Forest plot for pain outcome (visual analogue scale). Red boxes indicate studies with comparator exercise intervention equivalent to Pilates intervention.

**Figure 3 healthcare-11-01404-f003:**
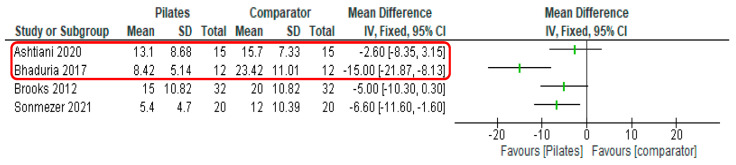
Forest plot for disability outcome (Oswestry low back pain disability questionnaire). Red box indicates studies with comparator exercise intervention equivalent to Pilates intervention.

**Table 1 healthcare-11-01404-t001:** Critical appraisal scores using the PEDro scale [64].

Study	Eligibility Specified	Random Allocation	Concealed Allocation	Groups Similar at Baseline	Participant Blinding	Therapist Blinding	Assessor Blinding	<15% Dropouts	Intention-to-Treat Analysis	Between-Group Difference	Point Estimate and Variability	Total
Ashtiani [58]	Y	1	0	1	0	0	0	1	0	1	1	5
Batibay [59]	Y	1	0	1	0	0	1	1	0	1	1	6
Bhadauria [56]	Y	1	0	0	0	0	0	0	0	1	1	3
Brooks [57]	Y	1	0	1	1	0	1	1	1	1	1	8
Cruz-Diaz [60]	Y	1	1	1	0	0	1	1	0	1	1	7
Mendes Tozim [61]	Y	1	0	1	0	0	0	1	0	1	1	5
Nabavi [62]	Y	1	1	0	0	0	1	1	0	1	1	6
Sonmezer [63]	Y	1	1	1	1	0	1	0	0	1	1	7

PEDro items to be scored where 1 = yes and 0 = no or not reported.

**Table 2 healthcare-11-01404-t002:** Study characteristics. Data are presented as mean (SD) unless otherwise specified.

Author Country Sample Size (Dropouts)	Age (Range)/Body Mass Index Gender	Intervention; Dosage; Setting; Practitioner	Relevant Outcome Domains	Assessment Method/ Muscles Assessed	Outcome Assessment Time Points
Pilates vs. equivalent exercise intervention					
Ashtiani (2020) [58] Iran 30 (0)	18–50 MP: 34.5 (6.4)/25.1 (3.9) GE: 35.5 (5.9)/26.5 (4.3) F: 30	MP: modified exercises; 3 times/week for 6 weeks (session length NR); University clinic (practitioner NR) GE: cycling, stretching, and strengthening; 3 times/week for 6 weeks (session length NR); University clinic (practitioner NR)	Muscle activation Pain Disability	Pressure biofeedback/ NR	NR
Bhadauria (2017) [56] India 44 (8)	20–60 MP: 35.3 (12.9)/26.0 (6.2) LS: 32.7 (11.7)/21.8 (2.9) DS: 36.7 (10.7)/24.7 (4.6) ‡ M: 24, ‡ F: 12	MP: 10 exercises; 60 min, 10 sessions over 3 weeks; Outpatient clinic, group of 12; Physiotherapist LS *: 16 exercises 10 s × 10 reps; 60 min, 10 sessions over 3 weeks; Outpatient clinic, group of 12; Physiotherapist DS: 14 exercises activating LES and RA 10 s × 10 reps; 60 min, 10 sessions over 3 weeks; Outpatient clinic, group of 12; Physiotherapist	Muscle activation Pain Disability	Pressure biofeedback/ NR	Baseline, 3 weeks
Mendes Tozim (2021) [61] Brazil 46 (5)	60–75 MP: 66.7 (3.6)/31.1 (8.0) GE: 68.5 (4.9)/26.2 (3.5) EG: 68.0 (4.7)/29.1 (5.3) F: 46	MP: 3–10 exercise variations increasing in number; fortnightly, 2 sets; 60 min, 2 times/week for 8 weeks; setting NR, group of 5; Physiotherapist Pilates instructor GE: walking, resistance training, stretching and balance exercises, fortnightly variations; 60 min, 2 times/week for 8 weeks; setting NR, group of 5; Physiotherapist EG *: 4 workshops; 30 min, fortnightly for 8 weeks; setting NR, group of 15; Physiotherapist	Muscle activation Pain	sEMG and lumbar dynamometer/ LM, IL	Baseline, 8 weeks
Nabavi (2018) [62] Iran 41 (0)	18–55 SE: 40.8(8.2)/24.9 (4.4) GE: 34.1 (10.8)/26.4 (3.2) NR	SE: 16 exercises 10 reps × 10 s; 3 times/week for 4 weeks (session length NR); University clinic, group size NR; Physiotherapist GE: 16 routine exercises 10 reps × 10 s; 3 times/week for 4 weeks (session length NR); University clinic, group size NR; Physiotherapist	Muscle thickness Pain	Real-time ultrasound/ TrA, LM	Baseline, 4 weeks
Pilates vs. non-equivalent exercise intervention					
Batibay (2021) [59] Turkey 60 (7)	18–60 MP: 49.3 (10.4)/25.0 (2.6) GE: 48.4 (9.3)/26.3 (2.7) F: 60	MP: basic/intermediate exercises 3 × 10 reps; 60 min 3 times/week for 8 weeks; Outpatient clinic, group of 8; Physiotherapist GE: general stretching and strengthening exercises 3 × 10 reps; 60 min; 3 times/week for 8 weeks; Home, individual; nil supervision	Muscle thickness Muscle activation Pain Disability Quality of life	Real-time ultrasound/ Right LM, TrA IO, EO, RA	Baseline, 8 weeks
Brooks (2012) [57] Australia 64 (12)	18–50 AP/MP: 36.2 (8.2)/NR GE: 36.3 (6.3)/NR M: 24, F: 40	AP/MP: combination of mat and reformer Pilates exercises; 50–60 min, 3 times/week for 8 weeks; setting NR, group of 10; supervisor with >5 years’ experience GE: indoor stationary cycling training; 50–60 min, 3 times/week over 8 weeks; setting NR, group of 10; supervisor with >5 years’ experience	Muscle activation Pain Disability	sEMG/TrA, IO, LES, RA	Baseline, 8 weeks
Pilates vs. non-exercise intervention					
Cruz-Diaz (2017) [60] Spain 102 (4)	18–50 MP: 36.9 (12.5)/NR AP: 35.5 (12.0)/NR Con: 36.3 (10.7)/NR ‡ M: 35, ‡ F: 63	MP †: 21 exercises; 50 min 2 times/week for 12 weeks; setting NR, group of 4; Physiotherapist Pilates instructor AP †: 14 reformer exercises; 50 min 2 times/week for 12 weeks; setting NR, group of 4; Physiotherapist Pilates instructor CG: no intervention	Muscle thickness Pain Disability	Real-time ultrasound/TrA	Baseline, 6 weeks, 12 weeks
Sonmezer (2021) [63] Turkey 50 (10)	20–35 MP: 29.0 (2.8)/23.8 (3.2) Con: 28.0 (2.1)/23.3 (2.6) F: 50	MP: 18 exercises, 2–3 sets of 3–12 reps progressed fortnightly; 60–70 min, 2 times/week for 8 weeks; setting NR, individual; Physiotherapist Pilates instructor CG: no exercise prescription. Usual prenatal care.	Muscle activation Pain Disability Quality of life	Pressure biofeedback/ TrA	Baseline, 8 weeks

AP: apparatus pilates group; CG: control group; DS: dynamic strengthening group; EG: educational group; EO: external oblique; F: female; GE: general exercise group; IL: iliocostalis lumborum; IO: internal oblique; LES: lumbar erector spinae; LM: lumbar multifidus; LS: lumbar strengthening group; M: male; MP: mat pilates group; MVIC: maximal voluntary isometric contraction; NR: not reported; RA: rectus abdominis; rep: repetitions; s: seconds; SE: stabilisation exercises; sEMG, surface electromyography; TENS: transcutaneous electrical nerve stimulator; TrA: transversus abdominis; * = exercises replicated in other intervention/comparator groups; † = three-arm trial with similar intervention groups combined for analysis; ‡ = participant gender reported after dropouts.

**Table 3 healthcare-11-01404-t003:** Summary of outcomes.

Study	Primary Outcomes				Secondary Outcomes					
	Muscle Thickness (Real-time Ultrasound)		Muscle Activation		Pain (Visual Analogue Scale)		Disability (Questionnaires)		Quality of Life (Questionnaires)	
	within Group	between Groups	within Group	between Groups	within Group	between Groups	within Group	between Groups	within Group	between Groups
Pilates vs. exercise intervention										
Ashtiani (2020) [58]			Biofeedback Bent knee fall out, Biering-Sorenson test ↑*+ Knee lift abdominal test ↑+	No difference	↓*+	↓*+	Oswestry ↓*+	Oswestry ND		
Bhadauria † (2017) [56]			Biofeedback ↑*+	Biofeedback ↑+	↓*+	↓+	Oswestry ↓*+	Oswestry ↓+		
Mendes-Tozim † (2021) [61]			Electromyography RMU ↑*+ LMU ↑+	Electromyography RMU ↑*+ LMU no difference	Not reported	Not reported				
Nabavi (2018) [62]	LLM, RLM, LTrA, RTrA ↑*+	LLM, RLM, LTrA, RTrA ↑+			↓*+	↓+				
Pilates vs. non-equivalent exercise intervention										
Batibay (2020) [59]	LM, TrA, IO ↑*+	LM, TrA, IO ↑*+	Sit-up test ↑*+	Sit-up test ↑+	↓*+	↓*+	BDI, QUB ↓*+	BDI, QUB ↓*+	SF-36 ↑*+	SF-36 ↑*+
Brooks (2012) [57]			Electromyography LTrA, IO ↓*+ RTrA, IO no difference	Electromyography Not reported	↓*+	↓*+	Oswestry ↓*+	Oswestry ↓*+		
Pilates vs. non exercise intervention										
Cruz-Diaz ‡ (2017) [60]	TrA ↑*+	TrA ↑*+			↓*+	↓+	RMDQ, TSK ↓*+	RMDQ, TSK ↓*+		
Sonmezer (2021) [63]			Biofeedback ↑*+	Biofeedback ↑*+	↓*+	↓*+	Oswestry ↓*+	Oswestry ↓*+	NHP ↑−	NHP ↓+

Key: ↑ = increase with intervention, ↓ = decrease with intervention; + = positive change; − = negative change; * = results are statistically significant (*p* < 0.05); † = multi-arm trial with an intervention group not relevant to review; ‡ = multi-arm trial with two pilates groups combined for data synthesis. Muscles: IO, internal oblique; LIO, left internal oblique; LLM, left lumbar multifidus; LM, lumbar multifidus; LTrA, left transverse abdominis; RLM, right lumbar multifidus; RIO, right internal oblique; RTrA, right transverse abdominis; TrA, transverse abdominal. Outcomes: (Disability) BDI, Beck Depression Inventory; QUB, Quebec low back pain and disability scale; RMDQ, Roland Morris Disability Questionnaire; TSK, Tampa Scale of Kinesiophobia. (Quality of life) NHP, Nottingham Health Profile; SF-36, Short form-36.

**Table 4 healthcare-11-01404-t004:** GRADE summary of findings.

		Certainty Assessment					No. of Participants		Impact	
Outcome	No. of Studies	Quality	Inconsistency	Indirectness	Imprecision	Other Considerations	Pilates	Control	Significance Reported	Certainty
Muscle activation (PBU)	2	fair *	serious †	serious ‡	very serious §	none	32	32	significant impact reported in one study	⨁◯◯◯ Very low
Muscle activation (sEMG)	2	good **	serious ††	low concern	very serious §	none	46	45	significant impact reported in one study	⨁◯◯◯ Very low
Muscle thickness (RTUS)	3	good **	serious †††	serious ‡	very serious §	none	118	81	significant impact reported in two studies	⨁◯◯◯ Very low
Pain (VAS)	8	fair *	serious ††††	serious ‡	very serious §	none	211	173	significant impact reported in four studies	⨁◯◯◯ Very low
Disability (ODI)	4	fair *	serious †††	serious ‡	very serious §	none	61	60	significant impact reported in two studies	⨁◯◯◯ Very low

* average PEDro score across studies 4–6; ** average PEDro score across studies 6–8; † between-study variance in outcomes may be explained by not all studies having equivalent interventions between control and experimental groups; †† only one study reported a between-groups analysis, unable to assess variance in outcomes; ††† between-study variance in outcomes may be explained by not all studies having equivalent interventions between control and experimental groups and one study not having similar groups at baseline; †††† between-study variance may be explained by baseline scores not equivalent for two studies and three studies not having equivalent interventions between control and intervention groups; ††† between-study variance in outcomes may be explained by not all studies having equivalent interventions between control and experimental groups and one study not having similar groups at baseline; ‡ indirectness of concern since some studies did not compare equivalent interventions, so the research question was different; § Low participant number, <400 participants; ⨁ score of certainty according to GRADE guidelines; ◯ does not meet GRADE guidelines for score.

## Data Availability

Data supporting reported results can be found within this manuscript.

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
