# Peer review of "Pilates to Improve Core Muscle Activation in Chronic Low Back Pain: A Systematic Review"

_healthcare, 2023, doi:10.3390/healthcare11101404_

Round 1

Reviewer 1 Report

The main question was on whether Pilates exercises could benefit people with chronic back pain.

It has some value for the Pilates instructors and the physicians treating people with chronic back pain. Overall, is somewhat original abut quite relevant to the field .

The problem is mainly the number of studies that are analysed as it is quite small to allow for valid conclusions to be drawn.

This is an interesting and well-designed systematic review, addressing an issue that will be of interest both to the physicians that treat people with chronic back pain and to the patients as well as the Pilates instructors. The main issue with this review, as the authors state themselves, is the fact that it was performed with only 8 studies. This greatly weakens the overall outcomes of their study and limits the impact of the review. Nevertheless, it can be used as a basic guide for future research designs for people willing to conduct similar RCT studies as well as for future systematic reviews on the subject.

In particular, there are some points in the text that they authors may consider and provide some changes:

Lines 14, 53 and elsewhere: RCT (probably refers to randomized control trials) should explain acronym as it is not mentioned in the abstract or in the main text.

Line 62: The search strategy was designed by the researchers and then it was reviewed by the independent librarian? Please clarify as this statement is confusing

Table 3: Nice idea to summarise the outcomes and compare the studies. It is though a little confusing with all the acronyms. Maybe there should be a better representation of the acronyms so it is easier to relate them to the table ? eg instead of having then at the caption maybe list them separately or maybe have two different tables; one with the primary and one with the secondary outcomes? I find it difficult to follow at the current presentation.

Author Response

Response to Reviewer 1 Comments

Point 1: This is an interesting and well-designed systematic review, addressing an issue that will be of interest both to the physicians that treat people with chronic back pain and to the patients as well as the Pilates instructors. The main issue with this review, as the authors state themselves, is the fact that it was performed with only 8 studies. This weakens the overall outcomes of their study and limits the impact of the review. Nevertheless, it can be used as a basic guide for future research designs for people willing to conduct similar RCT studies as well as for future systematic reviews on the subject.

Response 1: Thank you for this positive feedback. We have reinforced in the Limitations Section of the manuscript that the major weakness of this review is that only 8 studies were included in the final yield. This limits the generalisability of findings.

Point 2: Lines 14, 53 and elsewhere: RCT (probably refers to randomized control trials) should explain acronym as it is not mentioned in the abstract or in the main text. Yes we have explained this refers to Randomised Controlled Trial.

Line 62The search strategy was designed by the researchers and then it was reviewed by the independent librarian? Please clarify as this statement is confusing.  We have revised the manuscript for clarity, to say “A librarian worked with the research team to create and implement the search strategy.”

Table 3: Nice idea to summarize the outcomes and compare the studies. It is though a little confusing with all the acronyms. Maybe there should be a better representation of the acronyms, so it is easier to relate them to the table  eg instead of having then at the caption maybe list them separately or maybe have two different tables; one with the primary and one with the secondary outcomes? I find it difficult to follow at the current presentation. 

Response: Thank you for this feedback. We have removed as many abbreviations as possible, especially in Table 3 and made it clearer.

Reviewer 2 Report

A review completed to determine how pilates compared to alternate forms of exercise to improve muscle strength and reduce pain.

Seems both original and relevant as few studies were available.

Given the high prevalence of chronic back pain and the known correlation with muscular endurance, this would add to the body of literature in terms of exercise that patients are more likely to utilize, as many do not adhere to current guidelines.

The methodology could consider limiting the etiology of back pain in search but it will drastically limit the amount of articles found. May consider comparing to pharmacological treatments as an outcome.

The "aiuthors state in the conclusion that "For individuals wanting to improve core muscle strength and activation, it appears to be more beneficial than not exercising." And in the discussion that it "For people with chronic low back pain, Pilates was not inferior to equivalent exercise interventions to reduce pain. Nevertheless, it did appear to be more effective at reducing pain than not exercising." Could split the discussion into pain versus muscle strength to be more clear, as they are separate.

Overall this is a well written paper

Line 141- two trials of what?

RCT should probably be defined

Table 3 is a little hard to follow, if there was a way to make it more clear?

Author Response

Response to Reviewer 2 Comments

Point 1: A review completed to determine how Pilates compared to alternate forms of exercise to improve muscle strength and reduce pain. Seems both original and relevant as few studies were available. Given the high prevalence of chronic back pain and the known correlation with muscular endurance, this would add to the body of literature in terms of exercise that patients are more likely to utilize, as many do not adhere to current guidelines. The methodology could consider limiting the aetiology of back pain in search but it will drastically limit the amount of articles found. May consider comparing to pharmacological treatments as an outcome.

Response 1: Thankyou for this positive feedback. We decided not to limit further the aetiology of back pain in the search as this would probably have limited the yield of suitable articles even further.  We have added a new sentence in the Limitations section of the Discussion to say “We did not compare Pilates interventions with pharmacological treatments for chronic low back pain and this would be a valuable topic for future clinical trials.”

Point 2: The authors state in the conclusion that "For individuals wanting to improve core muscle strength and activation, it appears to be more beneficial than not exercising." And in the discussion that it "For people with chronic low back pain, Pilates was not inferior to equivalent exercise interventions to reduce pain. Nevertheless, it did appear to be more effective at reducing pain than not exercising." Could split the discussion into pain versus muscle strength to be more clear, as they are separate.

Response 2: This is a good suggestion thank you and we have separated out paragraphs in the Discussion to have one paragraph that focusses on muscle strength and the other paragraph to focus on Pain, as you have recommended.

Point 3: Overall this is a well written paper.  Thank you.

Line 141- two trials of what?  Thank you, we have now described the studies referenced.  Now fixed thanks

RCT should probably be defined  Yes we have defined this as randomised controlled trial.

Table 3 is a little hard to follow, if there was a way to make it more clear? Response: Thank you. One reason why it is less clear is all the abbreviations. So we have spelled them out where possible and clearly explained them in the footnote and made many minor changes for clarity.

Reviewer 3 Report

Introduction

It is not specified in the work what differences exist between therapeutic Pilates and other exercise practices. Why is Pilates considered therapeutic? At a legal level Pilates is a sport practice. The authors should better justify the use of this sport practice as a therapeutic tool.

Materials and Methods

It would be necessary to establish the PICO question in order to deepen the intervention proposals and justify the search criteria based on the proposed intervention as indicated in the PRISMA statement.

Line 97 Refers to the possibility of contacting the authors of the selected articles due to eligibility problems. It would be interesting to know if there was a need or changes in the selection of the articles selected for this reason.

The flowchart would be easier to read if it were on a single page.

Results

The tables of results for the pain and disability variables should show the p-values.

Conclusions

The conclusions drawn by the review are not very relevant and knowledgeable.

Any physical activity is better than doing nothing. But considering that we are dealing with a population with low back pain, other types of conclusions should be shown.

Author Response

Response to Reviewer 3 Comments

Point 1: It is not specified in the work what differences exist between therapeutic Pilates and other exercise practices. Why is Pilates considered therapeutic? At a legal level Pilates is a sport practice. The authors should better justify the use of this sport practice as a therapeutic tool.

Response 1: Thankyou for your review. We have edited the second paragraph in the introduction to further explain the therapeutic aspects of Pilates.

Regarding the reviewer's opinion that Pilates is a sports practice, we could find no evidence in the literature that it is solely used for sports people or that legally it is sports alone (maybe this differs across the globe?). There are many references to Pilates for medical conditions, pregnancy and a range of physical and non-motor symptoms. So we have respectfully not added new text re this.

Point 2: Materials and Methods. It would be necessary to establish the PICO question in order to deepen the intervention proposals and justify the search criteria based on the proposed intervention as indicated in the PRISMA statement.

Response: We agree and have added the PICO question as follows: The main aim of this systematic review was to synthesize and evaluate research conducted as randomised controlled trials (RCTs) to answer the question: Is Pilates exercise effective for improving core muscle strength / core muscle activation in people living with chronic low back pain? A secondary aim was to examine the effects of Pilates on pain, disability, and quality of life in people with chronic low back pain.

Line 97 Refers to the possibility of contacting the authors of the selected articles due to eligibility problems. It would be interesting to know if there was a need or changes in the selection of the articles selected for this reason.

Response: We have added that this was not necessary for the 8 articles reviewed.

The flowchart would be easier to read if it were on a single page. Yes, it is on one page.

Results The tables of results for the pain and disability variables should show the p-values.

Response: We do not want to make the Table even more complex, so we respectfully decline this suggestion as we have included this information in the footnote.

Conclusions Any physical activity is better than doing nothing. But considering that we are dealing with a population with low back pain, other types of conclusions should be shown.

Response We have considered this and changed the conclusion as follows: “There is emerging evidence that Pilates is not inferior to equivalently dosed exercises for improving core muscle strength in people living with chronic low back pain. The randomised controlled trials reviewed in this analysis provided some suggestion that Pilates is more beneficial than not exercising to improve core muscle strength and low back pain in some individuals”.

Reviewer 4 Report

This was a systematic review conducted to verify the impact of Pilates on improving muscle activation of core muscles in order to contrast and/or limit the occurrence on low back pain. The Physiotherapy Evidence Database scale (PEDro) was used to assess the methodological quality of the selected papers.

At the end of the research, authors isolated 8 RCT articles out of the 563 initially selected. These 8 articles were in line with the inclusion criteria previously defined (it is not clear what RCT means, as this acronym has not been explained). According to what found, authors concluded that Pilates can be used in alternative to equivalently dosed exercises, and can have a better effect compared to the absence of exercise for improving core muscle strength. Thus, to the authors’ opinion, Pilates can represent an effective intervention for treating people with chronic low back pain.

Although this is an interesting topic that deserves to be deeply investigated, to this reviewers’ opinion, there are a few major concerns that make this review and, most of all, the conclusions herein addressed, not completely acceptable and enough.

First of all, although this is mentioned at the end of the manuscript, authors wrote (line 376-378): “There is emerging evidence that Pilates is not inferior to equivalently dosed exercises for improving core muscle strength in people living with chronic low back pain. For individuals wanting to improve core muscle strength and activation, it appears to be more beneficial than not exercising”. Since authors started from 563 papers and reached the possibility to consider only 8 out of 563 articles, it does not seem to me that such a conclusion can be addressed only on 8 articles.

I’m quite sure that this conclusions could have been reached starting from a different selection of the papers.

In fact, and this is to me the major limitation of the review, I will try to highlight a few points that should be addressed and reconsidered very carefully.

Line 52-54. Authors wrote: “This systematic review aimed to synthesize and evaluate research conducted as RCTs, with a primary outcome of determining the effectiveness of Pilates exercises on core muscle strength or muscle activation in adults with chronic low back pain”. RCT stands for? It should be explained.

A major issue is related to the inclusion criteria initially defined. At line 80-81 authors wrote: “Studies were included if participants were adults aged 18 years and over who had low back pain of more than three months duration, with no exclusion criteria related to underlying aetiology”. Why didn’t authors consider exclusion criteria related to etiology? I think this is a mistake, low back pain related to a pathology such as a herniated disc is very different from non specific CLBP (cumulative low back pain). Putting them together is, to me, wrong. To me, it would be interesting to understand through the published works what is the different type of impact that Pilates could have also in relation to the different origin of Low Back Pain. Authors should expand on this point.

Author Response

Response to Reviewer 4 Comments

Point 1: This was a systematic review conducted to verify the impact of Pilates on improving muscle activation of core muscles in order to contrast and/or limit the occurrence on low back pain. The Physiotherapy Evidence Database scale (PEDro) was used to assess the methodological quality of the selected papers. At the end of the research, authors isolated 8 RCT articles out of the 563 initially selected. These 8 articles were in line with the inclusion criteria previously defined (it is not clear what RCT means, as this acronym has not been explained). According to what found, authors concluded that Pilates can be used in alternative to equivalently dosed exercises, and can have a better effect compared to the absence of exercise for improving core muscle strength. Thus, to the authors’ opinion, Pilates can represent an effective intervention for treating people with chronic low back pain.

Response: Thank you for these very positive comments.

Point 2: Although this is an interesting topic that deserves to be deeply investigated, to this reviewers’ opinion, there are a few major concerns that make this review and, most of all, the conclusions herein addressed, not completely acceptable and enough. First of all, although this is mentioned at the end of the manuscript, authors wrote (line 376-378): “There is emerging evidence that Pilates is not inferior to equivalently dosed exercises for improving core muscle strength in people living with chronic low back pain. For individuals wanting to improve core muscle strength and activation, it appears to be more beneficial than not exercising”. Since authors started from 563 papers and reached the possibility to consider only 8 out of 563 articles, it does not seem to me that such a conclusion can be addressed only on 8 articles.

Response: We can understand the reviewers concerns and therefore we have changed the conclusion as follows:

“There is emerging evidence that Pilates is not inferior to equivalently dosed exercises for improving core muscle strength in people living with chronic low back pain. The randomised controlled trials reviewed in this analysis provided some suggestion that Pilates is more beneficial than not exercising to improve core muscle strength and low back pain in some individuals”.

Point 3: I will try to highlight a few points that should be addressed and reconsidered very carefully.

Line 52-54. Authors wrote: “This systematic review aimed to synthesize and evaluate research conducted as RCTs, with a primary outcome of determining the effectiveness of Pilates exercises on core muscle strength or muscle activation in adults with chronic low back pain”. RCT stands for? It should be explained.

Response: Yes we have explained this refers to Randomised Controlled Trial.

Point 4 A major issue is related to the inclusion criteria initially defined. At line 80-81 authors wrote: “Studies were included if participants were adults aged 18 years and over who had low back pain of more than three months duration, with no exclusion criteria related to underlying aetiology”. Why didn’t authors consider exclusion criteria related to aetiology? I think this is a mistake, low back pain related to a pathology such as a herniated disc is very different from non specific CLBP (cumulative low back pain). Putting them together is, to me, wrong. To me, it would be interesting to understand through the published works what is the different type of impact that Pilates could have also in relation to the different origin of Low Back Pain. Authors should expand on this point.

Response We thank the reviewer for this suggestion and have added another section in the Limitations of the Discussion to say “In this review we did not consider exclusion criteria related to aetiology and it is possible that different results in response to Pilates could be found for different medical conditions”.

Round 2

Reviewer 3 Report

Thanks to the authors for taking into account many of the comments. I consider it a methodologically well done work that can contribute new lines of research on the subject.

Reviewer 4 Report

The manuscript in its new (revised) version has been improved according to the reviewer's comments.

Authors provided a detailed point by point answer and fixed the manuscript accordingly.

To this reviewer's opinion, no additional revision is required.